# Weiss or Wit: Chemical Profiling of Wheat Beers via NMR-Based Metabolomics

**DOI:** 10.3390/foods14091621

**Published:** 2025-05-03

**Authors:** Plamen Chorbadzhiev, Dessislava Gerginova, Svetlana Simova

**Affiliations:** 1Bulgarian NMR Centre, Institute of Organic Chemistry with Centre of Phytochemistry, Bulgarian Academy of Sciences, Acad. G. Bonchev Str. Bl. 9, 1113 Sofia, Bulgaria; plamen.chorbadzhiev@orgchm.bas.bg (P.C.); dessislava.gerginova@orgchm.bas.bg (D.G.); 2Faculty of Chemical and Systems Engineering, University of Chemical Technology and Metallurgy, 8 St. Kliment Okhridski Blvd., 1756 Sofia, Bulgaria; 3Centre of Competence “Sustainable Utilization of Bio-Resources and Waste of Medicinal and Aromatic Plants for Innovative Bioactive Products” (BIORESOURCES BG), 1000 Sofia, Bulgaria

**Keywords:** wheat beer, NMR metabolomics, chemometrics, craft and commercial beer

## Abstract

Wheat beers, including Witbier, Hefeweizen, and Weizenbock, are known for their unique sensory profiles, which are shaped by the combination of ingredients, fermentation conditions, and brewing methods. In this study, we used nuclear magnetic resonance (NMR) spectroscopy to explore the metabolomic signatures of various wheat beer styles and substyles. By analyzing 39 beer samples from 17 countries, we identified and quantified 50 metabolites, ranging from alcohols and saccharides to amino acids and organic acids. Ethanol and maltodextrin were the most abundant compounds, while higher contents of alcohols and organic acids played a key role in flavor variation. Through orthogonal partial least squares discriminant analysis (OPLS-DA), we achieved an impressive 97.44% accuracy in distinguishing between Witbier, Hefeweizen, and Weizenbock based on their metabolic profiles. The analysis also revealed notable compositional differences between craft and commercial beers, with craft beers showing higher concentrations of alcohols and amino acids. These results underscore the significant impact of raw materials, fermentation parameters, and brewing techniques on the chemistry of wheat beer. Furthermore, this study highlights the potential of NMR spectroscopy as a powerful tool for beer authentication and quality control.

## 1. Introduction

Beer has been identified as the world’s most popular alcoholic beverage, with a consumption rate that places it third among all beverages, after water and tea. In 2023, the global production of beer reached approximately 1.88 billion hectoliters, with a market value of approximately USD 1.1 trillion [1]. The origins of beer can be traced back to the Neolithic period, when grains such as barley, millet, and wheat were utilized in the brewing process [2,3]. By the 10th century, brewers had begun to enhance beer with various herbs and spices, which were later replaced by hops [4]. The German Purity Law (Reinheitsgebot) of 1516 established a framework for beer production, stipulating that beer could only contain barley, water, and hops, with wheat beer being an exception due to their status as a special privilege granted to certain families, such as the Degenbergers in Bavaria.

In recent years, consumer interest has shifted toward craft beer, produced by small, independent breweries that prioritize traditional methods, high-quality ingredients, and distinctive flavors over mass production [5]. Concurrent with this trend, wheat beer has witnessed a substantial rise in popularity. According to recent market analysis, the global wheat beer market was valued at approximately USD 5.3 billion in 2024 and is projected to reach approximately USD 8.2 billion by 2034. North America currently holds the largest market share (approximately 40%), followed by Europe (35%), and then Asia (20%) [6]. Throughout history, various wheat beer styles have emerged, with the most prominent being Belgian Witbier (“Wit”), German Hefeweizen (“Weiss”), and its substyles—Kristallweizen and Dunkelweizen—as well as the stronger Weizenbock. The diversity of wheat beers is notable, with styles reflecting their regional origins and brewing traditions. A prime example is Witbier, a traditional Belgian ale that is typically brewed with 40–60% unmalted wheat, malted barley, and, on occasion, oats. Its flavor profile is often enhanced by the addition of dried orange peel and coriander seeds, yielding floral and citrus notes. Witbier typically possesses an alcohol content ranging from 4.0% to 5.5% ABV (alcohol by volume). In contrast, German Hefeweizen is brewed with a minimum of 50% wheat malt and is known for its banana and clove aromas derived from distinctive hefeweizen yeast strains. This variety is characterized by its characteristic cloudiness, which is due to wheat proteins and residual yeast, and typically has ABVs ranging from 4.3% to 5.6%. The Hefeweizen substyles encompass Kristallweizen, a filtered version that yields a clear beer with a cleaner, crisper taste, and Dunkelweizen, a dark wheat beer brewed with roasted malts, providing a richer, maltier profile with caramel and toffee notes. The darker and stronger wheat beer variety, known as Weizenbock, is classified as an ale with an ABV range from 6.5% to 9.0% [7].

Notwithstanding their growing popularity and increasing scientific interest, a comprehensive characterization of the chemical composition of wheat beers remains elusive. The complexity of these beverages arises from the interplay of various ingredients and fermentation conditions, which makes it challenging to fully define their metabolite profiles. The majority of existing studies have centered on the analysis of volatile compounds, phenolics, fatty acids, carbohydrates, amino acids, organic acids, proteins, vitamins, elements, and polysaccharides. Research has primarily compared wheat and barley beers, as well as Witbier and Hefeweizen, often based on volatile organic compounds (VOCs). Other investigations have examined factors such as yeast strain, wheat type, hop variety, wheat concentration, raw material origin, fermentation conditions, coriander addition, the incorporation of fruits, and aging [8,9,10,11,12,13,14,15,16,17,18,19,20,21,22]. In the beer industry, authenticity and quality control are of paramount importance, particularly in the context of the growing popularity of craft brewing and premium wheat beer styles. Conventional analytical methods frequently lack the capacity to differentiate between craft and commercial beers, with the exception of certain cases that rely on sensory evaluation. The majority of wheat beer studies utilize gas chromatography–mass spectrometry (GC-MS) and high-performance liquid chromatography (HPLC), in conjunction with statistical analysis. However, a more comprehensive approach is necessary to fully capture the intricacies of the diverse array of wheat beer styles. In this context, nuclear magnetic resonance (NMR) spectroscopy has emerged as a highly promising analytical technique for beer characterization [23]. Its merits include the need for minimal sample preparation, the ability to simultaneously identify and quantify multiple metabolite classes without the need for calibration curves, and its suitability for both targeted and untargeted analyses. The structured and quantitative nature of NMR data, in conjunction with their low experimental variability, renders them well suited for multivariate chemometric methods such as principal component analysis (PCA) and orthogonal partial least squares discriminant analysis (OPLS-DA). These characteristics facilitate robust modeling and interpretation of metabolic patterns. NMR has been successfully used to differentiate ale and lager beers [24], craft and commercial beers [16], wheat and non-wheat beers [21,25], grape ales and wines [26], and beers from different production sites [27]. However, to date, no studies have yet compared the NMR profiles of different wheat beer styles, their substyles, or their craft and commercial origins. Despite its analytical advantages, NMR spectroscopy is not without its limitations in the field of metabolomics. These include lower sensitivity compared to mass spectrometry, which renders it less suitable for detecting low-abundance metabolites. Moreover, difficulties are encountered in the resolution of superimposed signals within complex mixtures. Furthermore, the considerable expense associated with instrumentation and maintenance has the potential to constrain the instrument’s extensive utilization.

In this study, NMR spectroscopy was used to characterize the chemical profiles of several wheat beer styles, including Hefeweizen, Weizenbock, Witbier, and the substyles of Hefeweizen—Kristallweizen and Dunkelweizen. A total of 50 compounds were identified and quantified, including alcohols, saccharides, amino acids, organic acids, nucleosides, nucleobases, and other metabolites. OPLS-DA was used to determine the characteristic compounds of each style. By providing a comprehensive chemical characterization, this research contributes to the understanding of wheat beer authenticity, quality control, and differentiation between styles and production methods. These insights have the potential to assist brewers, regulators, and consumers in evaluating and ensuring the integrity of various wheat beer styles.

## 2. Materials and Methods

### 2.1. Wheat Beer Samples

The present study analyzed 39 wheat beer samples representing five different styles: Dunkelweizen, Hefeweizen, Kristallweizen, Weizenbock, and Witbier. The samples were obtained from two retail sources: 33 samples were obtained from specialty craft markets, while 6 samples were obtained from standard supermarkets. The specific style, alcohol by volume (ABV), producer, and country of origin for each sample are listed in Appendix A. All samples were stored at 4 °C prior to the analysis.

### 2.2. Sample Preparation

Samples were prepared according to the methodology described in a previous study by Gerginova et al. [26]. The procedure involved degassing beer samples in an ultrasonic bath for 15 min. Subsequently, 500 μL of each degassed sample was combined with 50 μL of deuterated buffer (pH 4.4) containing 0.1% TSP (3-(trimethylsilyl)-2,2,3,3-tetradeuteropropionic acid sodium salt), 0.05% NaN_3_, and D_2_O. The deuterated phosphate buffer used for sample preparation had a pH of 4.4 and comprised 0.1 M KH_2_PO_4_ and H_3_PO_4_ in D_2_O, adjusted with NaOH. The prepared sample solutions were transferred to 5 mm NMR tubes for analysis.

### 2.3. NMR Analysis

^1^H NMR spectra were acquired using a Bruker Avance NEO 400 spectrometer, equipped with a triple resonance BBO probe head, operating at 300.0 ± 0.1 K. Water signal suppression was achieved using the zgcppr pulse sequence, with the following acquisition parameters: spectral width 13.2 ppm, 256 scans, 16 dummy scans, 42,104 data points, relaxation delay 4.0 s, and acquisition time 4.0 s. The resulting free induction decay (FID) was zero-filled and multiplied by a 0.3 Hz line broadening. Manual phase and multipoint baseline correction, suppressing protein signals, were applied using MestreNova 14.2.3. The TSP signal at 0 ppm was used as an internal reference. The identification of metabolites was accomplished through the integration of multiple approaches. The signals in the NMR spectra were initially matched to known compounds using private and public databases (HMDB, BMRB) and then compared to data in the literature. In order to confirm the presence of key metabolites, 2D NMR experiments (HSQC, TOCSY) were performed to resolve overlapping signals and provide additional structural information. In addition, a series of spike experiments were conducted using authentic standards for a selection of metabolites, including sugars, amino acids, and organic acids. These experiments were designed to verify the identities of these metabolites.

The quantification of all 50 unambiguously identified components was performed using a standard formula, which had been officially approved by the OIV (International Organisation of Vine and Wine) [28]. Due to signal overlap, deconvolution (line fitting) was used instead of standard integration. Appendix A shows an example of a deconvoluted spectrum.

### 2.4. Statistical Analysis

The quantitative data of the identified compounds were subjected to multivariate statistical analysis using orthogonal partial least squares discriminant analysis (OPLS-DA) and Nightingale’s diagrams. OPLS-DA is a supervised multivariate method employed for the classification of samples according to their chemical profiles. In contrast to standard partial least squares discriminant analysis (PLS-DA), which integrates predictive and orthogonal variation within a single component, OPLS-DA separates these components into distinct entities. The predictive component captures systematic variation related to class membership. Orthogonal components, on the other hand, are associated with variations that are not related to group differentiation. This separation enhances the interpretability of the model and mitigates noise, particularly in complex datasets such as NMR spectra, which contain both relevant and irrelevant signals. OPLS-DA is a widely utilized technique in metabolomics due to its capacity to elucidate group-related patterns and facilitate a more reliable interpretation of high-dimensional data.

OPLS-DA was performed using SIMCA 17.0 software (Umetrics, Umeå, Sweden), while Nightingale’s diagrams were generated in Microsoft Excel. OPLS-DA was used to visualize style separation and to identify key discriminatory compounds. Specifically, variable importance in projection (VIP) scores were applied to identify significant compounds for differentiating Hefeweizen, Witbier, and Weizenbock, as well as for distinguishing Kristallweizen, Dunkelweizen, and Hefeweizen. Nightingale’s diagrams were used to compare craft and commercial wheat beers. The performance of the classification models was assessed through misclassification tables (Appendix A) and receiver operating characteristic (ROC) curve analysis (Appendix A). To further evaluate the reliability of the models, 7-fold cross-validation was performed to determine the predictive accuracy. In addition, permutation testing with 25 iterations (Appendix A) was used to examine statistical significance and robustness.

## 3. Results and Discussion

### 3.1. NMR Profiling of Wheat Beer

In this study, we used ^1^H NMR spectroscopy to systematically identify and quantify a total of 50 metabolites in all samples of wheat beer. This approach provided comprehensive insights into the fermentation processes and sensory characteristics of the beverages. The metabolites were classified into several groups, with alcohols showing the highest concentrations, followed by sugars, organic acids, amino acids, nucleosides and nucleobases, and other minor compounds. Figure 1 presents a comparative ^1^H NMR spectrum with water suppression of three wheat beer styles, highlighting the main metabolite signals and the main differences in the concentrations of the compounds.

Additionally, the minimum, maximum, and average concentrations of the quantified compounds are presented in Table 1, along with the corresponding chemical shifts and multiplicities of the signals used for quantification.

Among the metabolites identified, ethanol and maltodextrin were the most abundant. Ethanol, the primary fermentation product, was the most abundant metabolite, ranging from 16,474 to 75,716 mg/L (average: 35,057 mg/L, corresponding to 4.4% ABV). These values are consistent with reports in the literature that Weizenbock has the highest ethanol content, followed by Weissbier and Witbier. The higher ethanol levels in Weizenbock are due to its higher malt content, longer fermentation, and yeast strain characteristics [7]. Glycerol, a fermentation by-product that enhances mouthfeel and viscosity, had the highest concentrations in Weizenbock (average: 3489 mg/L), followed by Hefeweizen (1915 mg/L) and Blanche (1604 mg/L). This distribution is indicative of differences in wort composition, yeast metabolism, and fermentation conditions. The range observed in this study (1172–3964 mg/L) is consistent with the results of previous studies, although it is slightly lower than the highest reported value of 4670 mg/L [29]. Methanol, primarily from pectin degradation, was detected at trace levels (1–18 mg/L), consistent with previously reported levels (0–27 mg/L) [30,31]. Higher levels of alcohols, major contributors to beer aroma, were quantified within the typical ranges: 1-propanol (9–50 mg/L), 2-phenylethanol (21–57 mg/L), isopentanol (isoamyl alcohol), and isobutanol [32]. Hefeweizen and Witbier showed values comparable to previous reports [10,33], while Weizenbock demonstrated elevated concentrations due to its different fermentation conditions and precursor availability. Weissbier contained higher levels of alcohols than Witbier, probably due to its unique yeast strain, and a higher wheat malt content. The formation of these compounds, primarily via the Ehrlich pathway, involves amino acids undergoing transamination, decarboxylation, and reduction. However, anabolic synthesis involving α-keto acids from carbohydrate metabolism may also contribute to their formation [11]. Their presence contributes fruity, floral, and spicy notes to the aroma of beer.

The saccharide composition showed considerable variation among the different categories of wheat beer. Maltodextrin, ranging from 2287 to 30,778 mg/L (average: 10,230 mg/L), contributes to beer body, residual sweetness, and stability by preventing excessive haze formation. The observed maltodextrin concentrations in this study exceeded the previously reported values for wheat beers (maximum: 21,490 mg/L) by Xu et al. and were lower than the values reported by Baiano et al. (maximum: 40,730 mg/L) [12,15]. These differences can be attributed to variations in mashing conditions, which affect starch degradation and enzymatic activity. Levels of xylose (average: 54 mg/L) and arabinose (33 mg/L), non-fermentable sugars derived from hemicellulose degradation, were lower than the values in the literature [15]. Mannose, a component of malt and yeast cell walls, exhibited a range of 6 to 59 mg/L, consistent with previous studies reported by Grieco et al. [29]. The presence of other saccharides, including kojibiose, raffinose, and αα-trehalose, was also observed, with their concentrations influenced by various factors such as raw materials, mashing protocols, and yeast activity. Sucrose, possibly derived from malt or adjuncts, was detected in several samples, whereas fructose was detected in only one sample (**wt7**). As expected, Weizenbock had the highest total saccharide content, while a comparison of Witbier and Weissbier revealed that Hefeweizen had more raffinose and kojibiose, while Witbier contained more maltodextrin and sucrose, likely due to the use of unmalted wheat and adjuncts.

A total of twelve organic acids, which play a key role in achieving optimal flavor balance and effective fermentation monitoring, were precisely quantified. Lactic acid, the most abundant acid, ranged from 45 to 1230 mg/L (average: 327 mg/L), exceeding previous reports by several authors, with the exception of Baiano’s results (680–1360 mg/L), which may reflect variations in hygiene or fermentation conditions [12,21,29]. Acetic acid, a by-product of yeast metabolism and a potential indicator of spoilage, ranged from 19 to 922 mg/L. This range is within the expected levels, as established by Xu’s findings, but is lower than those reported by Grieco et al. (1440–1890 mg/L) [15,29]. Succinic acid (0–419 mg/L), citric acid (0–201 mg/L), and malic acid (0–151 mg/L) contributed to sourness and overall balance, consistent with the values established in the literature [15]. Pyruvic acid (1–171 mg/L), a metabolic intermediate, was detected at moderate levels. GABA (8–206 mg/L), a neurotransmitter associated with yeast metabolism and potential health benefits, was detected at levels above 150 mg/L in samples **wb1**, **hw2**, and **hw13**. Gallic acid, a phenolic compound with antioxidant properties, was found in small amounts (6–20 mg/L), likely derived from malt, hops, or barrel aging in beers subjected to extended maturation. Its presence may contribute to oxidative stability and the prevention of lipid oxidation. Tartaric acid, with concentrations up to 45 mg/L, is likely derived from raw materials or adjuncts. The analysis revealed that formic acid (0–17 mg/L), associated with oxidation or microbial activity, was more abundant in darker wheat beers. Fumaric acid (0–10 mg/L) and maleic acid (1–20 mg/L), which are intermediates of the Krebs cycle, were also identified, but their concentrations were low and consistent with previously reported values [29]. As anticipated, Weizenbock samples had the highest total organic acid content, probably due to their higher original gravity and extensive fermentation. Witbiers followed, with elevated organic acid levels possibly due to the use of adjuncts such as orange peel and coriander seeds.

A total of nine amino acids were measured in the wheat beer samples, which are by-products of protein breakdown during the malting and fermentation processes. These amino acids act as essential nutrients for yeast, influencing the development of flavor and aroma. They also influence protein stability and foam persistence. The most abundant amino acid detected in all samples was pyroglutamic acid, with concentrations ranging from 24 to 314 mg/L. This cyclized derivative of glutamic acid was found at higher levels in dark wheat beers—Dunkelweizen and Weizenbock—likely due to the use of special darker malts that undergo more extensive protein modification during kilning. Pyroglutamic acid was identified as a contributor to the savory, umami-like flavors characteristic of these beers. Alanine, the second most abundant amino acid, was detected at concentrations ranging from 2 to 198 mg/L (average: 43 mg/L), which is close to the mean value reported by Kabelová et al. for lager beers (average: 40 mg/L) [14]. Alanine has been shown to play a role in the Maillard reaction and Strecker degradation, which contribute to beer color, maltiness, and mouthfeel. Tyrosine (average: 29 mg/L) and phenylalanine (average: 26 mg/L) are precursors for phenolic compounds that contribute to spicy and clove-like flavors, while tryptophan (average: 24 mg/L) is a precursor for indole compounds that impart subtle floral or earthy notes. Other amino acids, including valine (average: 21 mg/L), leucine (average: 19 mg/L), and isoleucine (average: 11 mg/L), are key precursors in the Ehrlich pathway, leading to the formation of higher levels of alcohols. The concentrations of these amino acids were within the range reported in the extant literature [14]. However, their concentrations varied among the three different beer styles. Weizenbock demonstrated higher levels of amino acids due to its higher original gravity and more extensive protein degradation during the mashing process. In contrast, Witbier, which contains unmalted wheat, had a reduced amino acid content compared to Hefeweizen, a beer that traditionally uses malted wheat.

The nucleosides adenosine (0–57 mg/L), inosine (0–18 mg/L), thymidine (2–35 mg/L), uridine (0–124 mg/L), and guanosine (6–88 mg/L) along with the nucleobase uracil (3–108 mg/L) were quantified in wheat beers. These compounds are by-products of RNA and DNA degradation during the malting and fermentation processes and serve as precursors for nucleotide synthesis and energy production. The levels of uridine observed in this study were consistent with those reported for lager beers [27]. Compared to grape ales, wheat beers showed higher concentrations of uridine and guanosine, suggesting differences in yeast metabolism, raw materials, and fermentation conditions [26]. Among the different wheat beer styles, Witbier had the highest adenosine levels, possibly due to differences in raw materials or mashing conditions. In contrast, Weizenbock had higher levels of inosine, thymidine, guanosine, and uracil, likely reflecting its higher original gravity and greater nucleic acid degradation.

A variety of minor compounds were identified in wheat beers, with each contributing to the chemical profile and sensory characteristics. Acetaldehyde (0–20 mg/L), a fermentation intermediate, was generally present at low levels, indicating efficient yeast metabolism, with only one sample exceeding 10 mg/L (**hw7**), which could impart a green apple off-flavor. Betaine (76–290 mg/L) and choline (30–252 mg/L), both derived from malt, play a role in yeast metabolism and stress tolerance, with choline levels exceeding 155 mg/L in all Weizenbock samples, possibly reflecting variations in raw materials. The concentration of HMF (0–7 mg/L), a by-product of the Maillard reaction associated with heat exposure during wort boiling, showed minimal variation in most samples, indicating that caramelization or aging effects were negligible. Isoamyl acetate (0–91 mg/L), a key ester responsible for fruity (banana-like) flavors, showed significant variation, likely due to differences in yeast strain. Trigonelline (1–9 mg/L), a Maillard precursor found in malt, was detected in all samples, further highlighting the biochemical complexity of wheat beer.

This comparison suggests that the distinct metabolomic and compositional profiles of the three primary wheat beer styles, Weizenbock, Weissbier, and Witbier, are driven by differences in raw materials, yeast metabolism, and fermentation conditions. While these observations are consistent with the existing literature, a more robust differentiation of these styles could be achieved through statistical analysis.

### 3.2. Differentiation of Hefeweizen, Weizenbock, and Witbier

In order to provide a comprehensive comparison of the chemical composition of Witbier, Hefeweizen, and Weizenbock, the quantitative data were subjected to OPLS-DA analysis. The three Weizenbock samples were designated as a separate class, while the fifteen Blanche samples were assigned to the Witbier group. The Hefeweizen group included 15 Weissbier samples along with 3 Dunkelweizen and 3 Kristallweizen samples. The VIP (variable importance in projection) score threshold was set at 0.86, and this analysis identified 37 metabolites that were key discriminators among the three groups. These included seven alcohols (1-propanol, 2-phenylethanol, 2,3-butanediol, ethanol, glycerol, isobutanol, isopentanol), five saccharides (arabinose, kojibiose, maltodextrin, mannose, raffinose), and eight organic acids (acetic, formic, fumaric, GABA, gallic, maleic, succinic, tartaric), along with eight amino acids (alanine, histidine, isoleucine, leucine, phenylalanine, pyroglutamic acid, tyrosine, valine), four nucleosides (guanosine, inosine, thymidine, uracil), and five other compounds (betaine, choline, isoamyl acetate, HMF, and trigonelline). The OPLS-DA model was developed based on these results using three predictive and six orthogonal components (R^2^ = 0.813, Q^2^ = 0.116).

The OPLS-DA score plot in Figure 2 shows the distinct composition of Weizenbock compared to the lighter wheat beers, confirming previous observations. This difference spans almost all of the compound classes, with Weizenbock containing higher concentrations of compounds due to factors such as the use of darker malts, higher boiling and fermentation temperatures, and a longer fermentation duration, nearly twice that of Weissbier and Witbier. The contribution plots in Appendix A reveal the significant compounds responsible for differentiating the three beer styles. The OPLS-DA analysis also confirmed that the Blanche beers contain higher levels of monosaccharides, likely due to the use of unmalted grains. The presence of these sugars can be attributed to the limited enzymatic degradation during the mashing process, as unmalted grains contain low levels of endogenous amylases. Furthermore, the analysis revealed the presence of elevated levels of certain amino acids (alanine, isoleucine, valine, tryptophan, and tyrosin), trigonelline, and gallic acid, suggesting the influence of raw materials, yeast metabolism, and thermal processing. The higher HMF levels observed in Witbier are likely due to the traditionally long and vigorous boiling used to extract flavors from spices, such as coriander and orange peel. In addition to the previously discussed elevated levels of certain alcohols, sugars, and nucleosides, Hefeweizen also showed higher concentrations of acetaldehyde and histidine. The misclassification table (Appendix A) indicates a model accuracy of 97.44%, with only one Weissbier sample (**hw4**) misclassified as a Witbier. This misclassification may be due to its commercial origin, where variations in ingredient ratios (wheat-to-barley) and brewing processes (yeast strain, fermentation parameters) are common. These findings underscore the need for further comparative analyses between craft and commercial wheat beers to improve our understanding of the underlying processes that shape beer chemistry.

### 3.3. Comparative Analysis of Craft and Commercial Wheat Beers

A detailed analysis was performed to elucidate the differences between commercial and craft wheat beers. To achieve this objective, both Witbier and Hefeweizen samples were divided into craft (*n* = 12) and commercial (*n* = 3) groups. This classification was based on criteria commonly accepted within the European Union, with a particular focus on production methods and ingredient quality. Craft beers are defined as those produced by independent breweries using traditional brewing techniques and high-quality raw materials. In contrast, commercial beers are produced in larger-scale industrial breweries that rely on standardized processes for mass production. Nightingale diagrams, as shown in Figure 3a,b, were used to clearly visualize the differences. Separate diagrams by compound class (alcohols, saccharides, organic acids, amino acids, nucleosides, and others) for both craft and commercial beer styles—Hefeweizen and Witbier—are presented in Appendix A.

As shown in the accompanying diagrams, the mean levels of most alcohols, with the exception of methanol, were found to be elevated in craft beers (both Hefeweizen and Witbier) compared to commercial beers. The average discrepancy was 13% for Weiss and 16% for Blanche, with the isobutanol levels being almost double in craft Witbier samples. Another notable observation is the increased content of amino acids, with the exception of histidine, in craft beers. This could be attributed to the use of a wider variety of yeast strains and warmer fermentation temperatures, which promote the production of higher levels of alcohols and influence the amino acid profile. Conversely, the concentrations of saccharides in craft beers tend to be lower, with the exception of raffinose. This may be because craft beers use a wider range of grains and adjuncts, which affect the types and amounts of saccharides, while commercial beers often use fewer complex sugars for cost and efficiency reasons. In addition, elevated levels of lactic acid, maleic acid, and succinic acid were observed in both craft beers, as reported by Palmioli et al. [16]. This is likely due to less controlled fermentation and possible contamination with lactic acid bacteria (LAB) that thrive in less regulated environments. Acetaldehyde and thymidine are also elevated, reflecting differences in brewing practices and fermentation conditions between craft and commercial beers. Other components differ between the two beer styles due to a variety of factors, as previously discussed in this study.

These compositional differences underscore the influence of brewing practices on beer chemistry, with craft methods promoting complex flavor development through microbial diversity and process variability. The commercial beer group in this study consists of only six samples, which may limit the generalizability and statistical significance of the comparison between craft and commercial wheat beers. This limitation in the sample size may introduce bias, rendering the results preliminary. In the interest of ensuring the highest standards of accuracy and reliability in future studies, it is recommended that the sample sizes in subsequent research be increased in terms of both size and representativeness. A broader comparison of different beer styles and substyles could provide deeper insights into the processes influencing beer chemistry. For this reason, a comparative analysis of the chemical profiles of Hefeweizen and its substyles, Dunkelweizen and Kristallweizen, was conducted.

### 3.4. Metabolomic Differences Between Dunkelweizen, Kristallweizen, and Hefeweizen

In order to observe the differences between Hefeweizen and its primary substyles, OPLS-DA was performed. A total of 21 German-style light beers were divided into three groups: Hefeweizen (n = 15), Dunkelweizen (n = 3), and Kristallweizen (n = 3). The analysis included nineteen compounds with VIP scores greater than 1, comprising five alcohols (2,3-butanediol, ethanol, glycerol, isobutanol, isopentanol), five saccharides (arabinose, kojibiose, maltodextrin, mannose, xylose), four organic acids (formic, maleic, succinic, tartaric), two amino acids (alanine, pyroglutamic acid), and betaine, choline, and HMF. The OPLS-DA model was built using three predictive and five orthogonal components, achieving a goodness of fit (R^2^) of 0.876, and a predictive value (Q^2^) of 0.271, indicating a satisfactory differentiation between the three groups.

The score plot (Figure 4) revealed significant differences between the Dunkelweizen group and the other two groups. The contribution plots in Appendix A show that Dunkelweizen beers had higher concentrations of almost all 19 compounds except succinic acid, ethanol, glycerol, and isopentanol. This discrepancy is likely due to the incorporation of darker malts, in addition to the conventional wheat malt. Conversely, Kristallweizen had the lowest concentrations of all components, a phenomenon that can be attributed to the filtration process it undergoes after fermentation, which removes some of the yeast and other soluble compounds. The misclassification table (Appendix A) further confirmed the robustness of the model, with an accuracy of 100%.

These results underscore the impact of raw material selection and processing techniques on the chemical characteristics of different wheat beer styles, providing valuable insights into their distinctive flavor and sensory profiles. However, it is important to note that the observed differences for Dunkelweizen and Kristallweizen are based on only three samples each. This limited sample size of this study hinders the statistical reliability of the chemometric models developed for these substyles. Therefore, the findings must be regarded as preliminary, and further studies employing larger datasets are recommended to validate and expand upon these observations. In view of the comparatively modest Q^2^ values in combination with the limited sample size, OPLS-DA models are employed for exploratory analysis with a view to avoiding an overinterpretation of their predictive ability.

## 4. Conclusions

This in-depth NMR-based metabolomic study offers crucial insights into the chemical profiles of various wheat beer styles and substyles. By analyzing 50 metabolites across multiple beer samples, we demonstrated the power of ^1^H NMR spectroscopy as a precise tool for beer characterization and quality control. Our findings highlighted significant compositional differences among Weizenbock, Hefeweizen, and Witbier, driven by variations in raw materials, particularly malt composition, adjunct use, yeast strains, and fermentation conditions. OPLS-DA further confirmed these distinctions, with Weizenbock exhibiting higher metabolite levels, Witbier showing more sugars from adjuncts, and Hefeweizen standing out with unique esters and amino acids. A notable discovery was the distinct metabolomic profiles of craft versus commercial beers, emphasizing the role of brewing practices in terms of flavor and quality. These results reinforce the potential of NMR spectroscopy as a valuable tool for beer authentication, process optimization, and style differentiation. Expanding this approach to explore more wheat beer substyles and production methods could deepen our understanding of beer chemistry, support innovation in brewing science, and assist with regulatory compliance. Ultimately, this research offers valuable knowledge for brewers, regulators, and researchers striving to enhance beer differentiation and guarantee product authenticity.

## Figures and Tables

**Figure 1 foods-14-01621-f001:**
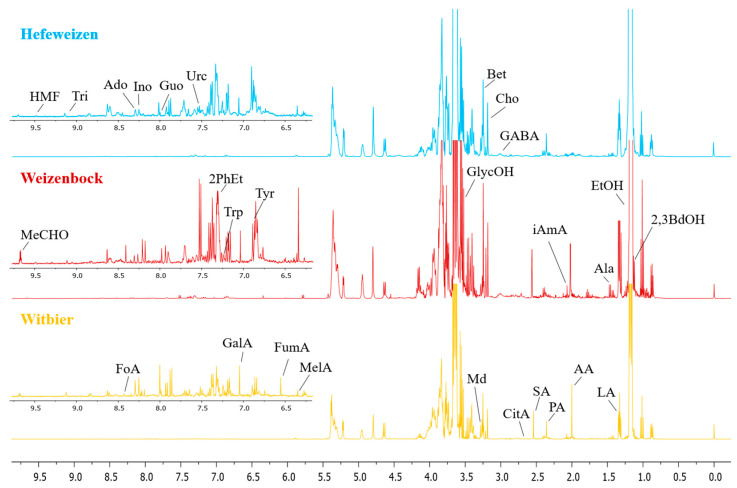
Comparative ^1^H NMR spectra of Hefeweizen (**hw15**, blue), Weizenbock (**wb3**, red), and Witbier (**wt10**, yellow), highlighting key metabolite signals: AA—acetic acid, Ado—adenosine, Ala—alanine, Bet—betaine, Cho—choline, CitA—citric acid, EtOH—ethanol, FoA—formic acid, FumA—fumaric acid, GABA—gamma-aminobutyric acid, GalA—gallic acid, GlycOH—glycerol, Guo—guanosine, HMF—5-hydroxymethylfurfural, iAmA—isoamyl acetate, Ino—inosine, LA—lactic acid, Md—maltodextrin, MeCHO—acetaldehyde, MelA—maleic acid, PA—pyruvic acid, SA—succinic acid, Tri—trigonelline, Trp—tryptophan, Tyr—tyrosine, 2,3BdOH—2,3-butanediol, 2PhEt—2-phenylethanol, and Urc—uracil.

**Figure 2 foods-14-01621-f002:**
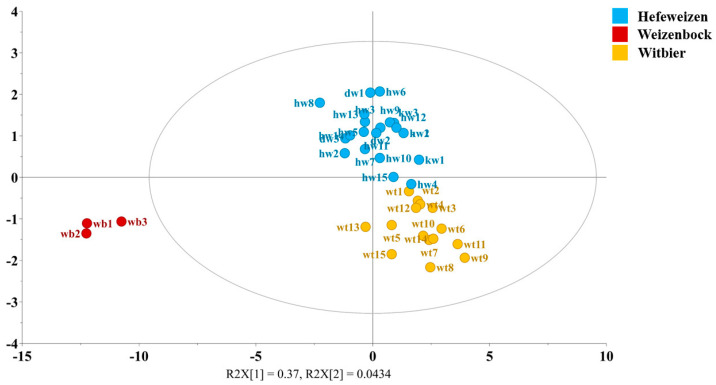
OPLS-DA score plot illustrating compositional differences between the three classes: Hefeweizen (blue), Weizenbock (red), and Witbier (yellow).

**Figure 3 foods-14-01621-f003:**
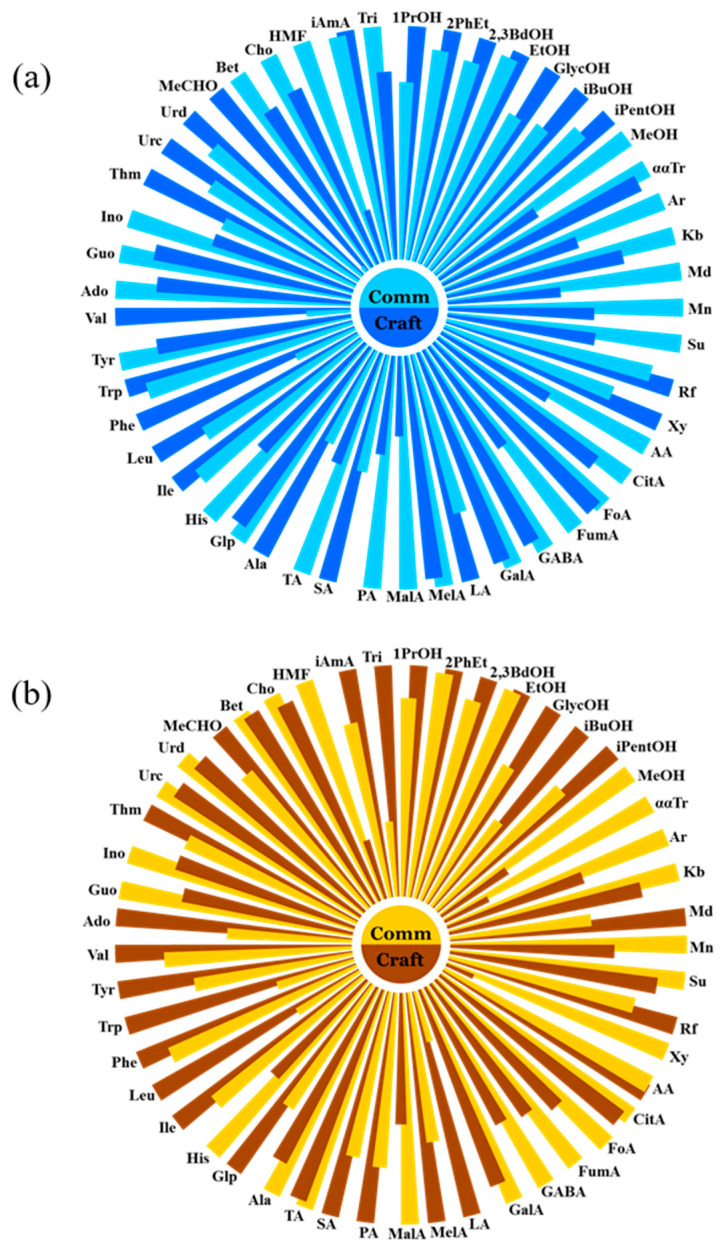
Nightingale diagrams comparing metabolite levels in commercial (Comm) and craft (Craft) samples of (**a**) Hefeweizen and (**b**) Witbier.

**Figure 4 foods-14-01621-f004:**
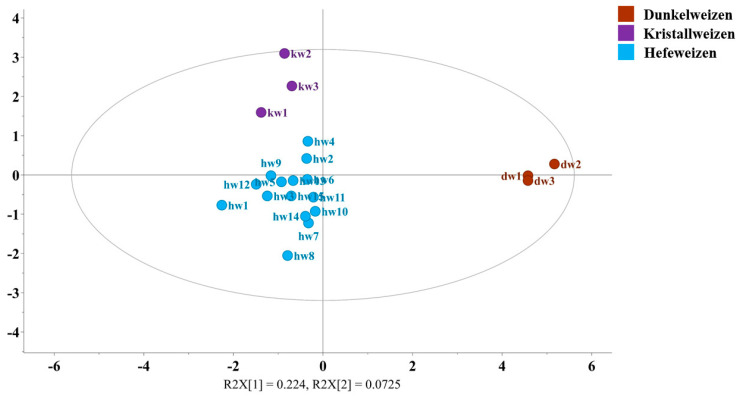
OPLS-DA score plot illustrating significant compositional differences between Dunkelweizen (brown), Kristallweizen (purple), and Hefeweizen (blue).

**Table 1 foods-14-01621-t001:** Concentrations (minimum, maximum, average) of quantified metabolites and corresponding ¹H NMR data (chemical shifts, signal multiplicities) for the studied wheat beer types.

	Weissbier (Hefeweizen)	Weizenbock	Witbier (Blanche)	^1^H δ in ppm
	min–max *average*	min–max *average*	min–max *average*	(multiplicities)
** *Alcohols* **				
1-propanol (1PrOH)	9.3–50.1 *24.8*	22.7–38.2 *32.9*	9.6–22.2 *15.6*	1.54 (m)
2-phenylethanol (2PhEt)	18.8–64.7 *41.6*	36.9–134.1 *73.0*	21.4–57.2 *30.4*	7.31 (m)
2,3-butanediol (2,3BdOH)	56.2–256.5 *132.0*	120.0–237.7 *187.7*	74.4–138.4 *101.0*	1.13 (d)
Ethanol (EtOH)	16,473.6–60,133.4 *33,840.4*	62,325.3–75,715.9 *67,943.6*	22,692.9–40,044.5 *30,184.2*	1.17 (t)
Glycerol (GlycOH)	1172.2–3600.7 *1915.7*	3167.8–3963.8 *3489.1*	1233.6–2745.4 *1663.8*	3.55 (dd)
Isobutanol (iBuOH)	27.7–103.2 *58.9*	58.3–235.0 *134.9*	9.8–59.2 *31.3*	1.74 (m)
Isopentanol (iPentOH)	26.4–123.7 *62.4*	88.9–159.2 *120.8*	34.4–88.4 *51.5*	1.43 (q)
Methanol (MeOH)	1.7–18.3 *4.8*	4.9–11.3 *7.5*	0.5–15.9 *3.5*	3.35 (s)
** *Saccharides* **				
αα-Trehalose (ααTr)	2.2–101.9 *21.3*	12.4–56.4 *29.9*	6.1–95.3 *21.4*	5.18 (d)
Arabinose (Ar)	6.4–61.6 *31.8*	25.5–68.8 *46.8*	16.6–50.2 *32.7*	4.49 (d)
Fructose (Fr)	-	-	0.0–1724.0 *114.9*	4.03 (d)
Kojibiose (Kb)	42.6–274.1 *164.4*	117.9–619.1 *370.7*	27.3–306.0 *136.3*	5.09 (d)
Maltodextrin (Md)	2287.5–16,466.7 *8592.7*	11,073.9–30,777.6 *21,468.3*	4948.8–21,812.3 *10,275.7*	3.26 (dd)
Mannose (Mn)	5.9–55.9 *23.3*	23.0–59.1 *46.2*	10.6–36.2 *25.2*	5.17 (d)
Sucrose (Su)	12.0–253.3 *96.5*	49.3–157.5 *105.7*	22.1–238.7 *113.0*	4.20 (d)
Raffinose (Rf)	21.5–149.1 *65.4*	75.5–144.0 *115.2*	12.8–105.8 *34.7*	5.02 (d)
Xylose (Xy)	5.9–216.1 *50.4*	29.7–170.1 *89.7*	5.2–205.1 *51.5*	5.17 (d)
** *Organic acids* **				
Acetic acid (AA)	29.5–715.5 *207.6*	423.7–922.5 *653.9*	18.6–286.6 *106.8*	2.00 (s)
Citric acid (CitA)	0.0–201.8 *108.6*	73.8–196.0 *123.8*	0.0–144.8 *82.5*	2.65 (d)
Formic acid (FoA)	0.0–17.5 *3.5*	4.0–13.2 *7.6*	0.6–4.9 *2.3*	8.42 (s)
Fumaric acid (FumA)	0.5–6.6 *2.8*	0.5–6.4 *4.2*	0.5–8.6 *3.4*	6.54 (s)
GABA	15.7–161.6 *65.6*	76.7–205.7 *143.2*	7.6–104.3 *37.6*	3.02 (t)
Gallic acid (GalA)	6.4–13.2 *9.7*	10.5–20.0 *14.4*	6.2–15.4 *10.0*	7.04 (s)
Lactic acid (LA)	44.8–1229.9 *334.6*	108.8–963.9 *476.7*	66.2–568.6 *285.3*	1.33 (d)
Maleic acid (MelA)	0.8–9.2 *2.8*	4.3–19.9 *9.7*	1.2–5.0 *2.2*	6.35 (s)
Malic acid (MalA)	0.0–150.8 *66.6*	41.6–106.7 *74.4*	0.0–201.5 *78.4*	2.76 (dd)
Pyruvic acid (PA)	11.4–170.7 *59.9*	10.0–107.4 *44.4*	1.4–114.3 *54.2*	2.36 (s)
Succinic acid (SA)	0.0–418.7 *136.7*	141.2–360.9 *231.3*	43.8–196.3 *113.2*	2.54 (s)
Tartaric acid (TA)	1.2–17.8 *5.4*	7.6–45.1 *22.4*	1.9–13.4 *5.0*	4.56 (s)
** *Amino acids* **				
Alanine (Ala)	2.4–85.6 *31.9*	115.2–198.3 *167.4*	4.4–148.1 *33.3*	1.47 (d)
Histidine (His)	0.0–4.9 *2.4*	1.2–6.9 *3.7*	0.8–3.5 *1.7*	7.84 (s)
Isoleucine (Ile)	1.0–21.3 *6.9*	19.0–68.9 *49.4*	1.0–20.5 *8.4*	1.00 (d)
Leucine (Leu)	1.0–36.8 *15.5*	56.4–115.5 *89.0*	1.5–42.6 *10.8*	0.95 (d)
Phenylalanine (Phe)	0.7–65.4 *21.7*	74.8–141.0 *108.7*	0.0–70.0 *16.2*	7.42 (m)
Pyroglutamic acid (Glp)	56.9–311.2 *139.8*	214.4–313.7 *277.3*	24.0–130.8 *83.5*	2.49 (m)
Tryptophan (Trp)	6.6–40.9 *21.2*	5.7–63.8 *36.0*	6.7–82.2 *24.7*	7.25 (t)
Tyrosine (Tyr)	12.2–43.7 *22.6*	41.1–104.1 *81.6*	6.4–52.0 *26.3*	6.88 (d)
Valine (Val)	0.7–48.7 *11.3*	97.5–152.4 *116.3*	1.0–79.4 *16.3*	0.98 (d)
** *Nucleosides and nucleobases* **				
Adenosine (Ado)	0.0–29.5 *14.9*	0.5–20.0 *8.0*	0.0–56.7 *18.4*	8.36 (s)
Guanosine (Guo)	7.4–62.6 *24.6*	35.7–87.6 *63.3*	6.1–30.7 *15.2*	7.98 (s)
Inosine (Ino)	0.0–17.7 *8.3*	5.1–15.4 *10.1*	0.0–14.2 *4.9*	8.34 (s)
Thymidine (Thm)	3.1–34.7 *12.6*	7.3–32.9 *22.5*	2.4–8.1 *4.6*	7.65 (s)
Uracil (Urc)	8.0–57.9 *21.8*	36.2–107.6 *63.2*	2.6–37.1 *13.3*	7.53 (d)
Uridine (Urd)	0.0–79.6 *45.3*	18.3–123.7 *75.2*	0.0–83.9 *42.6*	7.87 (d)
** *Other* **				
Acetaldehyde (MeCHO)	0.2–20.5 *2.7*	0.5–7.4 *2.9*	0.0–4.8 *1.2*	9.67 (q)
Betaine (Bet)	75.8–244.3 *143.6*	220.6–290.5 *259.0*	92.0–210.7 *134.1*	3.25 (s)
Choline (Cho)	29.6–151.0 *92.2*	159.8–252.3 *196.3*	31.4–117.5 *66.2*	3.19 (s)
Hydroxymethylfurfural (HMF)	0.0–2.4 *0.7*	0.3–1.6 *0.9*	0.5–7.3 *1.4*	9.45 (s)
Isoamyl acetate (iAmA)	10.7–91.4 *39.3*	57.8–86.2 *75.1*	0.0–65.6 *25.2*	2.07 (s)
Trigonelline (Tri)	1.0–7.0 *3.2*	4.3–8.8 *6.4*	1.3–9.1 *4.0*	9.11 (s)

## Data Availability

The original contributions presented in this study are included in the article/Appendix A. Further inquiries can be directed to the corresponding author.

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
