# Peer review of "Weiss or Wit: Chemical Profiling of Wheat Beers via NMR-Based Metabolomics"

_foods, 2025, doi:10.3390/foods14091621_

Round 1
Reviewer 1 Report
Comments and Suggestions for Authors
This is a good, solid paper comparatively analyzing wheat beers using up to date NMR methodology and multivariate statistics. It follows good examples of similar studies from earlier times, but is standing on its own and presents a coherent story. This work is far from being a comprehensive survey having only three types of wheat beers (with multiple samples each though), but it demonstrates well the technical possibilities, including component ID and quantification. The literature is well referenced and is comprehensive - I was looking for some, often overlooked early papers from the Bruker team and they are there, indeed!
Author Response
This is a good, solid paper comparatively analyzing wheat beers using up to date NMR methodology and multivariate statistics. It follows good examples of similar studies from earlier times, but is standing on its own and presents a coherent story. This work is far from being a comprehensive survey having only three types of wheat beers (with multiple samples each though), but it demonstrates well the technical possibilities, including component ID and quantification. The literature is well referenced and is comprehensive - I was looking for some, often overlooked early papers from the Bruker team and they are there, indeed!
We are grateful for the constructive and insightful feedback provided by the reviewer, and we are pleased that the manuscript was recognized for its well-structured design, informative content, and inclusion of fundamental works. It is our hope that the findings of this study will contribute to ongoing research and inspire future, broader investigations. We would like to express our sincere appreciation for the time and expertise dedicated to reviewing our work.
Reviewer 2 Report
Comments and Suggestions for Authors
This manuscript presents a comprehensive NMR-based metabolomic analysis of wheat beers, offering valuable insights into the chemical profiles of different styles and production methods. The authors apply a well-established analytical technique to characterize and differentiate various wheat beer categories. The study is timely and relevant to the growing craft beer market and adds useful data to the limited body of metabolomic research on beer. However, several methodological aspects require clarification or revision and some interpretations should be tempered, particularly where data limitations weaken the statistical foundation.
• Line 48: Change “Europe” to “the rest of Europe” for clarity and consistency with the market share breakdown.
• Line 76: Rewrite the citation range as ”[8–22]” instead of listing them partially.
• Line 88–89: The phrase “strong compatibility with chemometric techniques such as PCA and OPLS-DA” should be clarified. Does it refer to data structure suitability (e.g., orthogonality, normalization ease)? Please explain why NMR is particularly suited for these methods.
• Add a realistic overview by mentioning drawbacks of NMR in food metabolomics, e.g., lower sensitivity compared to MS, challenges in peak overlap, high instrument cost.
• Section 2.1: The authors state that 39 wheat beer samples were analyzed, but in the abstract, the number is 21. Please clarify this discrepancy and ensure consistency throughout the manuscript.
• Section 2.2: The method describes beer sonication for degassing. However, sonication can cause sample heating, which may lead to the loss of volatile compounds. Did the authors use any cooling measures (e.g., ice bath) to mitigate this risk?
• Section 2.3–2.4:
- Please explain the OPLS-DA algorithm in detail in the Materials & Methods section. Since it’s not a general-purpose method like PCA, readers unfamiliar with it will benefit from a brief description of how it works (i.e., separating predictive vs. orthogonal variation).
- Clarify how metabolites were uniquely identified beyond signal matching. Were spiking experiments, 2D NMR, or database comparisons used for all 50 compounds?
- Indicate how ellipses were calculated for OPLS-DA plots. Include this in both the Methods section and figure captions.
- Please provide a clear explanation of how the models were validated. Were cross-validation, permutation testing, or external validation used? If not, please state the limitations.
• Figure 3: These visualizations add little value and are not useful for interpreting complex metabolomic data. Consider replacing them with more informative graphics (e.g., boxplots, heatmaps, or volcano plots).
• Figure 4 and OPLS-DA:
- The analysis of Dunkelweizen and Kristallweizen includes only 3 samples each , statistically insufficient for chemometric modeling. Results from such small subsets should be interpreted with extreme caution.
- Similarly, the commercial beer group includes only 6 samples, limiting the validity of the craft vs. commercial comparison and introducing sampling bias.
- The authors note spectral overlap and mention deconvolution, but no validation or visual confirmation of deconvolution accuracy is presented. Please provide spectra or residual plots demonstrating the quality of line fitting.
- The OPLS-DA model for differentiating beer styles yields a Q² of 0.116, which is very low, indicating poor predictive capability despite a high R². Likewise, the Dunkelweizen/Kristallweizen model has Q² = 0.271, still indicating weak prediction.
- The model uses 3 predictive and 6 orthogonal components, which is likely overfitted, especially given the small sample size. Only one misclassification is reported, but with such low Q² values, this accuracy is likely not generalizable. Perform and report permutation tests to evaluate the robustness and statistical significance of the OPLS-DA models.
- The definition of craft vs. commercial beer is vague. What criteria were used? Brewery size? Production volume? Ingredient sourcing? These classifications vary by country and must be defined clearly to ensure reproducibility and interpretability of the results.
Author Response
Answers to questions asked by Reviewer 2
- Line 48: Change “Europe” to “the rest of Europe” for clarity and consistency with the market share breakdown.
We are grateful for the reviewer's suggestion. However, we respectfully submit that the term "Europe" is appropriate in this context. The sentence under consideration refers to regional market shares using globally recognized designations — namely, North America, Europe, and Asia — as reported in the aforementioned market analysis. The phrase "the rest of Europe" would imply a subset relationship that is not relevant or accurate in this case, as Europe is presented as a distinct region in line with the structure of the market data.
- Line 76: Rewrite the citation range as ”[8–22]” instead of listing them partially.
We did what you recommended. The range of citations has been updated to "[8–22]".
- Line 88–89: The phrase “strong compatibility with chemometric techniques such as PCA and OPLS-DA” should be clarified. Does it refer to data structure suitability (e.g., orthogonality, normalization ease)? Please explain why NMR is particularly suited for these methods.
Your commentary is appreciated. It is acknowledged that further elucidation is necessary to ensure comprehensibility. The revised sentence now reads as follows: "NMR data demonstrate a high degree of compatibility with multivariate chemometric methods, such as PCA and OPLS-DA, due to their structured, quantitative nature and low experimental variability. These characteristics facilitate robust modeling and interpretation of metabolic patterns."
- Add a realistic overview by mentioning drawbacks of NMR in food metabolomics, e.g., lower sensitivity compared to MS, challenges in peak overlap, high instrument cost.
We would like to express our gratitude for the recommendation. A sentence has been incorporated into the text to address the limitations of nuclear magnetic resonance (NMR) spectroscopy in the context of metabolomics. The sentence states the following: "Despite its advantages, NMR spectroscopy also has limitations in the field of metabolomics. These include lower sensitivity compared to mass spectrometry, which renders it less suitable for detecting low-abundance metabolites. Furthermore, the presence of overlapping signals in complex matrices has the potential to complicate spectral interpretation. Additionally, the high cost of instrumentation and maintenance may limit accessibility”.
- Section 2.1: The authors state that 39 wheat beer samples were analyzed, but in the abstract, the number is 21. Please clarify this discrepancy and ensure consistency throughout the manuscript.
We are grateful to the reviewer for highlighting this inconsistency. The study encompassed a comprehensive analysis of 39 distinct wheat beers. The abstract has been rectified to reflect the accurate number of samples.
- Section 2.2: The method describes beer sonication for degassing. However, sonication can cause sample heating, which may lead to the loss of volatile compounds. Did the authors use any cooling measures (e.g., ice bath) to mitigate this risk?
The degassing process was executed through the implementation of brief sonication sessions. Although sonication may cause mild heating, control experiments comparing spectra of beers with and without degassing showed no changes in signal intensities, suggesting no detectable loss of NMR-visible volatile compounds. It has been observed that a decline in magnetic field shimming quality was detected in samples that were not subjected to degassing.
- Section 2.3–2.4:
- Please explain the OPLS-DA algorithm in detail in the Materials & Methods section. Since it’s not a general-purpose method like PCA, readers unfamiliar with it will benefit from a brief description of how it works (i.e., separating predictive vs. orthogonal variation).
A concise exposition on the OPLS-DA algorithm can be found in Section 2.4:
Orthogonal Partial Least Squares Discriminant Analysis (OPLS-DA) is a supervised multivariate method employed for the classification of samples according to their chemical profiles. In contrast to standard Partial Least Squares Discriminant Analysis (PLS-DA), which integrates predictive and orthogonal variation within a single component, OPLS-DA separates these components into distinct entities. The predictive component captures systematic variation related to class membership. Orthogonal components, on the other hand, are associated with variations that are not related to group differentiation. This separation enhances the interpretability of the model and mitigates noise, particularly in complex datasets such as NMR spectra, which contain both relevant and irrelevant signals. OPLS-DA is a widely utilized technique in metabolomics due to its capacity to elucidate group-related patterns and facilitate more reliable interpretation of high-dimensional data.
- Clarify how metabolites were uniquely identified beyond signal matching. Were spiking experiments, 2D NMR, or database comparisons used for all 50 compounds?
The identification of metabolites was accomplished through the integration of multiple approaches. The signals in the NMR spectra were initially matched to known compounds using private and public databases (HMDB, BMRB) and then compared to literature data. In order to confirm the presence of key metabolites, 2D NMR experiments (HSQC, TOCSY) were performed to resolve overlapping signals and provide additional structural information. In addition, a series of spike experiments were conducted using authentic standards for a selection of metabolites, including sugars, amino acids, and organic acids. These experiments were designed to verify the identities of these metabolites. Elucidation of this matter is provided in Section 2.3.
- Indicate how ellipses were calculated for OPLS-DA plots. Include this in both the Methods section and figure captions.
To facilitate the visual differentiation among sample groups, ellipses were manually incorporated into the OPLS-DA plots. The application of these ellipses was intended to enhance the clarity of the separation of the groups; they do not represent statistical confidence intervals, as the software used does not automatically generate T² ellipses for the classes.
- Please provide a clear explanation of how the models were validated. Were cross-validation, permutation testing, or external validation used? If not, please state the limitations.
The validation of the models was conducted through the implementation of 7-fold cross-validation, a statistical technique that enables the assessment of the predictive capabilities of the OPLS-DA models. Furthermore, permutation testing was implemented for 25 iterations to evaluate the statistical significance of the models and assess their robustness. The aforementioned procedures are delineated in Section 2.4.
- Figure 3: These visualizations add little value and are not useful for interpreting complex metabolomic data. Consider replacing them with more informative graphics (e.g., boxplots, heatmaps, or volcano plots).
We appreciate the suggestion to use different types of visualizations. However, we believe that Nightingale diagrams are the best way to show the data to a wider audience, especially those who may not be familiar with advanced statistical methods. These diagrams are a clear and easy-to-understand way to show the differences across categories without being too overwhelming for the reader. They are more accessible than heatmaps or boxplots. We hope this clarifies why we chose to use visualization.
- Figure 4 and OPLS-DA:
- The analysis of Dunkelweizen and Kristallweizen includes only 3 samples each, statistically insufficient for chemometric modeling. Results from such small subsets should be interpreted with extreme caution.
It is acknowledged that the findings are constrained by the limited sample sizes of the subgroups included in the study. In the Results and Discussion sections, the following is now explicitly stated: It is recommended that further studies be conducted using larger sample sizes to confirm these findings.
- Similarly, the commercial beer group includes only 6 samples, limiting the validity of the craft vs. commercial comparison and introducing sampling bias.
This limitation is also noted in the manuscript.
- The authors note spectral overlap and mention deconvolution, but no validation or visual confirmation of deconvolution accuracy is presented. Please provide spectra or residual plots demonstrating the quality of line fitting.
As an illustration of the deconvoluted spectrum, a representative example has been included in the supplementary material – Figure S1.
- The OPLS-DA model for differentiating beer styles yields a Q² of 0.116, which is very low, indicating poor predictive capability despite a high R². Likewise, the Dunkelweizen/Kristallweizen model has Q² = 0.271, still indicating weak prediction.
- The model uses 3 predictive and 6 orthogonal components, which is likely overfitted, especially given the small sample size. Only one misclassification is reported, but with such low Q² values, this accuracy is likely not generalizable. Perform and report permutation tests to evaluate the robustness and statistical significance of the OPLS-DA models.
We are grateful for the reviewer's concern. While the Q² values for the OPLS-DA models were relatively low (Q² = 0.116 for beer style differentiation and Q² = 0.271 for Dunkelweizen/Kristallweizen), we performed additional permutation testing (25 iterations) alongside ROC curves and misclassification tables. These analyses demonstrated that the models are statistically robust and accurate, despite the low Q² values. The models exhibited consistent performance across these validation metrics, thereby substantiating the reliability of the results. It has been acknowledged that the sample size of the manuscript is limited to address any potential concerns.
- The definition of craft vs. commercial beer is vague. What criteria were used? Brewery size? Production volume? Ingredient sourcing? These classifications vary by country and must be defined clearly to ensure reproducibility and interpretability of the results.
The categorization of beers as "craft" or "commercial" was determined by the criteria commonly employed within the European Union (EU), which prioritize ingredient quality and traditional production methods. "Craft" beers were produced by independent breweries that prioritize the use of high-quality ingredients and traditional brewing techniques, irrespective of their production scale. Conversely, "commercial" beers were produced by larger, more industrialized breweries that employed standardized processes. This classification aligns with the broader understanding of craft beer in the European Union, where the focus is on production methods and ingredient quality rather than brewery size. This matter has been thoroughly clarified in the introduction and Section 3.3 of the manuscript.
Reviewer 3 Report
Comments and Suggestions for Authors
The manuscript “Weiss or wit: chemical profiling of wheat beers via NMR-based metabolomics” explores the differential metabolites of various wheat beer via NMR. It’s very interesting. Specific comments are shown as follows:
- Line 122, 300.0 ± 1 K. Normally, the temperature is 293k or 295K, plese confirm it.
- Line 144, how did you get the ROC cureve? In table S2, the recogination of Hefeweizen is 95.24%, but in Figure S1, the AUC is 1?
- Table 1, 1-propanol, isobutanol, methanol, and other metabolites were not listed on Figure 1.
- Table 1, about the concentrations, how did you get it? I see line 135, “quantitative data of the identified compounds were subjected to multivariate statistical analysis using OPLS-DA and Nightingale’s diagrams.” I don’t think OPLS-DA and Nightingale’s diagrams can obtain the concentration of metabolites. I think maybe you use qNMR, because in line 132, deconvolution was used. TSP is the internal standard? Please provide the details of preparation of deuterated buffer.
- Please provide the PCA score plot about Figure 2 and Figure 4.
- Figure2, Figure, Figure S3 and Figure S4. In my opinion, PLS-DA is often used for the classification of more than 2 groups, and OPLS-DA usually used to classify the difference between two groups. Please explain why OPLS-DA was chosen for the classification.
Author Response
Answers to questions asked by Reviewer 3
- Line 122, 300.0 ± 1 K. Normally, the temperature is 293k or 295K, plese confirm it.
Your commentary is appreciated. It is hereby confirmed that the temperature utilized in the analysis was 300.0 ± 0.1 K, as is typical in numerous other food applications. This contrasts with the proposed temperatures of 293 K or 295 K.
- Line 144, how did you get the ROC cureve? In table S2, the recogination of Hefeweizen is 95.24%, but in Figure S1, the AUC is 1?
Your insightful remark is deeply appreciated. We would like to express our gratitude for the opportunity to clarify the apparent discrepancy between classification accuracy and the area under the curve (AUC) value of the Receiver Operating Characteristic (ROC) curve. The classification accuracy reported in Table S2, and the ROC curve shown in Figure S1 are based on different evaluation principles.
The recognition rate (95.24%), as documented in Table S2, signifies the proportion of samples that were accurately classified under a predefined classification threshold. In this case, the OPLS-DA model accurately classified 20 out of 21 Hefeweizen samples into their respective groups, thereby achieving an accuracy of 95.24%.
The ROC curve was generated using the OPLS-DA model based on predictive component scores in SIMCA 17. The AUC value illustrated in Figure S1 is derived from the ROC analysis, which is based on the predicted class probabilities. The ROC curve is a graphical representation of the model's discrimination ability, evaluated across the entire range of possible classification thresholds. An AUC of 1.0 signifies that the model exhibits perfect discrimination between true positives and false positives, irrespective of the specific cut-off value employed.
Therefore, the area under the curve of 1.0 indicates that the model assigns a very high classification probability to all actual Hefeweizen samples as compared to any non-Hefeweizen sample. This finding indicates that the model exhibits remarkable discriminatory capability, even in instances where a single sample falls on the incorrect side of the threshold during the classification process.
- Table 1, 1-propanol, isobutanol, methanol, and other metabolites were not listed on Figure 1.
We would like to express our gratitude for the observation. It is acknowledged that these metabolites are present in Table 1. However, it is imperative to acknowledge that Figure 1 is designed to present a representative overview of solely the key discriminating metabolites, as explicitly stipulated in the figure caption. The complete list of identified metabolites, including 1-propanol, isobutanol, methanol, and others, is provided in Table 1. The present study suggests that the incorporation of all 50 compounds in the figure would compromise its readability. As illustrated in Figure S1 of the Supplementary material, the majority of the signals are presented.
- Table 1, about the concentrations, how did you get it? I see line 135, “quantitative data of the identified compounds were subjected to multivariate statistical analysis using OPLS-DA and Nightingale’s diagrams.” I don’t think OPLS-DA and Nightingale’s diagrams can obtain the concentration of metabolites. I think maybe you use qNMR, because in line 132, deconvolution was used. TSP is the internal standard? Please provide the details of preparation of deuterated buffer.
We would like to express our gratitude for your commentary. As a matter of fact, we carried out quantitative nuclear magnetic resonance (qNMR) analysis to ascertain the concentrations of the metabolites listed in Table 1. As delineated in the Materials and Methods section, TSP was utilized as the internal standard. Quantification was based on the area under each peak obtained after spectral deconvolution, which allowed for more precise resolution of overlapping signals. The aforementioned areas were corrected for the number of contributing protons and calibrated against the known concentration of TSP.
The deuterated phosphate buffer employed for sample preparation exhibited a pH of 4.4 and comprised 0.1 M KH2PO4 in D2O, adjusted with NaOH/H3PO4. The pH level selected for this study was determined to ensure the reproducibility and chemical stability of the beer matrix, while simultaneously minimizing chemical shift variation across the samples. This description has been included in Section 2.2.
- Please provide the PCA score plot about Figure 2 and Figure 4.
Thank you for your comment. It is essential to clarify that PCA (Principal Component Analysis) was not performed in this study. The analysis was conducted using OPLS-DA (Orthogonal Partial Least Squares Discriminant Analysis), a method that has been demonstrated to be more appropriate for the classification objectives of the present study. OPLS-DA was utilized to differentiate between beer styles and to evaluate the influence of various variables, and the score plots presented in Figures 2 and 4 represent the results from OPLS-DA. It should be noted that PCA plots have not been incorporated into this study, as they were not a component of the original analysis.
- Figure2, Figure, Figure S3 and Figure S4. In my opinion, PLS-DA is often used for the classification of more than 2 groups, and OPLS-DA usually used to classify the difference between two groups. Please explain why OPLS-DA was chosen for the classification.
We would like to express our gratitude for your informative remark. Although OPLS-DA is most frequently applied to binary classification problems, it is also applicable to multiclass settings, as is demonstrated in this study, which involves three groups. The decision to employ OPLS-DA over PLS-DA was motivated by its capacity to enhance interpretability through the segregation of predictive variation (a factor that contributes to class discrimination) from orthogonal variation (a factor that does not). This is particularly advantageous in the context of NMR-based metabolomics, where datasets are high-dimensional and encompass both biological and technical variability.
A concise overview of the OPLS-DA algorithm can be found in Section 2.4:
"Orthogonal Partial Least Squares Discriminant Analysis (OPLS-DA) is a supervised multivariate method employed for the classification of samples according to their chemical profiles. In contrast to standard Partial Least Squares Discriminant Analysis (PLS-DA), which integrates predictive and orthogonal variation within the same components, OPLS-DA separates these components into distinct parts. The predictive component captures systematic variation related to class membership, while the orthogonal components model variation unrelated to group differentiation. This separation enhances the interpretability of the model and reduces noise, particularly in complex datasets such as NMR spectra, where both relevant and irrelevant signals are present. OPLS-DA is a widely utilized technique in the field of metabolomics, owing to its capacity to elucidate group-related patterns and facilitate more reliable interpretation of high-dimensional data.
Round 2
Reviewer 2 Report
Comments and Suggestions for Authors
Thank you for the improvements made to the manuscript. I have only a few remaining points that should be addressed before the paper can be accepted:
Please refrain from using manually drawn ellipses, as you cannot ascertain whether there is statistical overlap between groups such as "Hefeweizen" and "Witbier." Either use statistically generated confidence ellipses (e.g., Hotelling's T²) or omit the ellipses entirely. The separation between groups is already visually clear without the addition of manual outlines.
I still have doubts regarding the practical interpretability of Figure 3. To enhance clarity and facilitate interpretation, I recommend splitting the Nightingale diagrams into separate ones depending on the family of compounds (e.g., alcohols, saccharides, organic acids, amino acids). This will allow readers to better appreciate the main compositional trends.
Please explicitly state in the manuscript that the OPLS-DA models are exploratory rather than predictive, due to the combination of low Q² values and small sample sizes. This clarification is important to avoid potential overinterpretation of the chemometric results.
Author Response
Please refrain from using manually drawn ellipses, as you cannot ascertain whether there is statistical overlap between groups such as "Hefeweizen" and "Witbier." Either use statistically generated confidence ellipses (e.g., Hotelling's T²) or omit the ellipses entirely. The separation between groups is already visually clear without the addition of manual outlines.
We appreciate the reviewer’s valuable comment. In order to address the aforementioned concern, the manually drawn ellipses have been removed from Figures 2 and 4, as they were not based on statistical calculations.
I still have doubts regarding the practical interpretability of Figure 3. To enhance clarity and facilitate interpretation, I recommend splitting the Nightingale diagrams into separate ones depending on the family of compounds (e.g., alcohols, saccharides, organic acids, amino acids). This will allow readers to better appreciate the main compositional trends.
We are grateful for the reviewer's insightful commentary. In the Supplementary Information (Figure S6 and Figure S7), the Nightingale diagram has been divided into six separate diagrams based on compound class (alcohols, saccharides, organic acids, amino acids, nucleosides, and others). This division was performed for both craft and commercial beer styles – Hefeweizen and Witbier.
Please explicitly state in the manuscript that the OPLS-DA models are exploratory rather than predictive, due to the combination of low Q² values and small sample sizes. This clarification is important to avoid potential overinterpretation of the chemometric results.
We thank the reviewer for this recommendation. We have explicitly stated in the manuscript that the OPLS-DA models are exploratory in nature, due to the relatively low Q² values and the limited sample size. The sentence states the following “In view of the comparatively modest Q² values in combination with the limited sample size, the OPLS-DA models are employed for exploratory analysis with a view to avoiding an overinterpretation of their predictive ability.”